# HDAC Inhibition to Prime Immune Checkpoint Inhibitors

**DOI:** 10.3390/cancers14010066

**Published:** 2021-12-23

**Authors:** Edith Borcoman, Maud Kamal, Grégoire Marret, Celia Dupain, Zahra Castel-Ajgal, Christophe Le Tourneau

**Affiliations:** 1Department of Drug Development and Innovation (D3i), Institut Curie, 75005 Paris, France; edith.borcoman@curie.fr (E.B.); maud.kamal@curie.fr (M.K.); gregoire.marret@curie.fr (G.M.); celia.dupain@curie.fr (C.D.); zahra.castelajgal@curie.fr (Z.C.-A.); 2INSERM U900 Research Unit, 92210 Saint-Cloud, France; 3Faculty of Medicine, Paris-Saclay University, 75005 Paris, France

**Keywords:** immunotherapy, epigenetics, HDAC inhibitor, combination therapies

## Abstract

**Simple Summary:**

Immunotherapy has made a breakthrough in medical oncology with the approval of several immune checkpoint inhibitors in multiple cancer types. Since only a minority of patients experience a durable response with these agents, combination strategies and novel immunotherapy drugs were developed to further counteract tumor immune escape. Epigenetic regulations can be altered in oncogenesis, favoring tumor progression. The development of epidrugs has allowed for successfully targeting these altered epigenetic patterns in lymphoma and leukemia patients. It has been recently shown that epigenetic alterations can also play a key role in tumor immune escape. Epidrugs, like HDAC inhibitors, can prime the anti-tumor immune response, therefore constituting interesting partners to develop combination strategies with immunotherapy agents.

**Abstract:**

Immunotherapy has made a breakthrough in medical oncology with the approval of several immune checkpoint inhibitors in clinical routine, improving overall survival of advanced cancer patients with refractory disease. However only a minority of patients experience a durable response with these agents, which has led to the development of combination strategies and novel immunotherapy drugs to further counteract tumor immune escape. Epigenetic regulations can be altered in oncogenesis, favoring tumor progression. The development of epidrugs has allowed targeting successfully these altered epigenetic patterns in lymphoma and leukemia patients. It has been recently shown that epigenetic alterations can also play a key role in tumor immune escape. Epidrugs, like HDAC inhibitors, can prime the anti-tumor immune response, therefore constituting interesting partners to develop combination strategies with immunotherapy agents. In this review, we will discuss epigenetic regulations involved in oncogenesis and immune escape and describe the clinical development of combining HDAC inhibitors with immunotherapies.

## 1. Introduction

The oncogenesis is a result of a multi-step process that allows malignant cells to progressively acquire specific abilities, like resistance to apoptosis, genomic instability with accumulation of mutations, inactivation of tumor suppressor genes, for example, that all belong to the so-called hallmarks of cancer cells [1]. Historically, treatments in medical oncology have been tailored according to the primary tumor location and histological type. The improvement and accessibility of technologies towards molecular profiling have allowed a considerable evolution with the emergence of precision medicine. Understanding and identifying the molecular alterations in oncogenesis have allowed the developing of targeted therapies based on identified molecular alterations. At the beginning in specific tumor types, such as trastuzumab in HER2-positive breast or gastric cancers [2,3], and, more recently, in a histology-independent way, like larotrectinib or entrectinib. Theses latter drugs target the *Neutrophic Tyrosine Receptor Kinase* (*NTRK*) gene fusion and were approved in an agnostic way in patients with cancers harboring an *NTRK* gene fusion, as this molecular alteration is shared among patients with different cancer types [4,5].

One important hallmark of cancer is the ability acquired by cancer cells to avoid immune destruction [1]. Understanding immune evasion that led to the development of immunotherapies has been a breakthrough in oncology, starting with the approval of immune checkpoint inhibitors (ICI) targeting cytotoxic T-lymphocyte-associated antigen 4 (CTLA-4) and programmed cell death 1 (PD-1) or its ligand programmed death-ligand 1 (PD-L1) to restore anti-tumor immune response [6]. ICI have been shown to improve overall survival (OS) in multiple cancer types. However, only a minority of patients experience durable responses to these agents and a huge effort is made to understand mechanisms of resistance to immunotherapies.

Beside genomic alterations, growing evidences have enlightened the role of epigenetic changes in the regulation of gene expression, like modifications of DNA methylation or changes in chromatin structure via modifications in histone acetylation, for example. Epigenetic modifications are, by definition, not directly coded by the DNA sequence, and, unlike genetic alterations, are usually reversible, especially using pharmacological agents. These epigenetic patterns can be altered in oncogenesis, favoring tumor cells development and progression, and have been investigated as new promising drug targets for cancer treatment via drugs targeting DNA methylation or histone deacetylation. Additionally, epigenetic alterations can play a role in immune evasion and tumor resistance to ICI.

In this review, we will describe the main epigenetic patterns involved in cancer and immune resistance, and we will further review the preclinical and clinical data supporting the rationale for combining epidrugs with ICI for the treatment of solid tumors, with a focus on inhibitors of histone deacetylases (HDACs).

## 2. Overview of Epigenetic Alterations in Cancer

### 2.1. Epigenetic Regulations

Epigenetic regulations comprise post-synthesis chemical modifications of DNA, RNA, and proteins, that are essential for the control and adaptability of biological processes [7]. Epigenetic modifications are reversible and dynamically regulated, explaining the highly dynamic nature of the epigenome. These mechanisms control the three-dimensional conformation of the chromatin, the localization and activity of RNA binding proteins, so as to dynamically fine-tune gene expression. Epigenetics dysregulations have been found to be implicated in many diseases and can especially drive to aberrant transcriptional programs that can promote cancer onset and progression. We will further describe the role of key epigenetic processes, DNA methylation, and post-translational modification of histones, as well as the dysregulation of these epigenome processes implicated in cancer development.

### 2.2. DNA Methylation

DNA methylation is a highly conserved biological regulatory process widespread in mammalian genomes, deregulations of methylation pathways being linked to many diseases [7]. The DNA methylation involves an alkylation reaction whereby a methyl group replaces a hydrogen atom, catalyzed by DNA methyltransferases (three known enzymes in human: DNMT1, DNMT3A, and DNMT3B) which use S-adenosylmethionine (SAM) as the methyl donor. This reaction occurs predominantly in palindromic CpG dinucleotides of DNA, adding a methyl group to the 5′ position of the cytosine pyrimidine ring [7]. The methylation process is regulated during the cell cycle by DNA demethylases (“erasers”), DNMTs (“writer”), and DNA methyl “readers”. CpG DNA methylation is a fundamental mechanism which is enriched in heterochromatin and in inactive gene promoters to maintain a repression state, and also in repetitive elements to prevent their transcription, while CpG islands (CpG-rich regions) are usually unmethylated in promoter regions of active genes.

In cancer, the first epigenetic alteration observed has been a global hypomethylation state that can promote genomic instability and activation of oncogenes [8]. Notably, many tumor suppressor genes are silenced by DNA methylation in cancer cells during the oncogenesis process [9]. Oncogenesis has also been associated with specific mutations in gene implicated in methylation machinery, like IDH1 or IDH2 mutations in acute myeloid leukemia (AML) or gliomas leading to aberrant DNA methylation patterns at genes involved in proliferation and differentiation, and gains of histone methylation [10].

These observations have led to the development of the first DNMT inhibitors used in clinic, like azacitidine or decitabine, which have broad cellular effects leading to global loss of DNA methylation, and have been approved for the treatment of myelodysplastic syndrome and AML [11,12,13]. At higher dose, these drugs will also induce DNA damage and cytotoxicity by direct incorporation of these cytidine analogues into RNA and DNA. However, DNMT inhibitors have shown limited clinical activity as monotherapy for the treatment of solid tumors, and are currently assessed in combinations with other oncologic treatments, including chemotherapy, targeted therapies, or immunotherapies [7].

### 2.3. Histone Modifications

Histone proteins represent dynamic components of the chromatin structure and play a critical role in the regulation of gene transcription by modification of the chromatin accessibility to the transcription machinery. Histone acetylation consists in the addition of an acetyl group to histone proteins to the lysine residues within the N-terminal tail, catalyzed by enzymes known as histone acetyltransferases (HATs) [14]. The acetylation of histones at specific positions (H3K9 residues, which mean at lysine 9 of histone 3, for example) leads to a transcriptionally active chromatin structure (euchromatin), due to the neutralization of the positive charge by the acetylation of lysine residues, diminishing their ability to bind with the negatively charged DNA [15]. Conversely, deacetylation of these residues, catalyzed by HDACs, leads to an inactive, condensed chromatin structure (heterochromatin). HDACs not only affect gene expression through the modulation of histone tail modifications, but have also been implicated in the regulation of non-histone proteins, including transcription factors, signal transduction mediators, DNA repair enzymes, chaperone proteins, as well as nuclear import regulators and cytoplasmic proteins [16]. In human, there are 18 HDAC enzymes that are divided into four classes, based on sequence similarities: Class I (HDACs 1, 2, 3, and 8) are mainly located in the nucleus, whereas Class IIa (HDACs 4, 5, 7, and 9) and IIb (HDACs 6 and 10) enzymes are located both in the nucleus and the cytoplasm, suggesting a cytoplasmic functional role for Class II HDACs [16]. Class III contains members of the sirtuins family, and also have non-histone substrates. Class IV contains only one poorly studied member: HDAC11.

Several correlative data have shown that HDACs are often overexpressed in various type of cancers, conferring a poor prognosis [14]. In addition, HDACs can be aberrantly recruited at specific gene promoters by oncogenic fusion proteins that drive leukaemogenesis. As an example, the AML1-ETO fusion protein found in patients with t (8; 21) AML is recruiting several HDACs enzymes that will repress AML1 target genes, thus preventing myeloid differentiation and inducing cellular transformation [17].

Histone methylation can also be disrupted in cancer, resulting in changes in gene expression and genome integrity, for example, affecting the gain of H3K27me3, a known gene-repressive histone modification [7]. Histone methylation dysregulation can also occur through various mechanism like mutations in gene encoding for enhancer of zeste homologue 2 (EZH2), the first methyltransferase responsible for the H3K27me3 histone modification, for example [18], or in some cases through oncogenic driver mutations directly in histone genes, like in glioblastomas, and especially pediatric glioblastomas in which around 30% contain mutations in histone genes [19].

These observations have led to the development of HDAC inhibitors as therapeutic drugs in cancer, at the beginning in hematologic malignancies. Vorinostat targeting multiple HDACs was the first HDAC inhibitor approved for treatment of refractory cutaneous T-cell lymphoma [20], followed by romidepsin [21] and belinostat [22] in the same setting.

Pan-HDAC inhibitors (inhibiting HDACs class I, II, and IV) lead to global hyperacetylation of histone and non-histone HDAC substrates. This can induce a range of cellular and molecular effects, including antitumor effect, such as tumor cell death, cell cycle arrest, senescence, differentiation, and increased tumor immunogenicity highlighting the function of HDAC inhibitors as anti-cancer drugs [23]. Several new molecules have been developed, like benzamide derivatives, that are inhibiting more specifically class I HDACs and, more recently, other isoform-selective HDAC inhibitors [23]. Many clinical trials are actually assessing HDACs inhibitors in various types of tumors with a focus on hematological malignancies, either as single agents or in combination with standard chemotherapies, targeted therapies, other epidrugs and, more recently, immunotherapies.

Other epidrugs have also been developed, like inhibitors of different histone methyltransferases, such as EZH2 inhibitors [24]. It is important to underline that epigenetic regulators might have a better effect in a disease-specific context, for example, with increasing response rates of EZH2 inhibitors in tumors with EZH2-activating mutations [24].

We will further discuss the rational development of HDAC inhibitors in the setting of solid tumors.

## 3. Rationale for Combining an HDAC Inhibitor with Immunotherapy in Oncology

### 3.1. Immune Checkpoints Inhibition

T-cell activation requires the engagement of the T-cell receptor (TCR) via MCH-peptide antigen complex presentation on antigen-presenting cells (APC), but other costimulatory signals are essential to an effective activation, like the interaction of the CD28 costimulatory signal on T cells with B7 molecules (CD80/CD86) that are predominantly expressed on APC [6]. Upon T-cell activation, T cells can proliferate and produce cytokines. To prevent damages to normal cells, T cells further upregulate inhibitory molecules that attenuate T-cell activation. For example, CTLA-4 is an inhibitory molecule that attenuates T-cell activation, expressed immediately following TCR engagement on T cells, bearing structural homology to CD28, that binds to B7 molecules with a higher affinity than CD28 [6]. By developing anti-CTLA-4 antibodies, the aim was to remove the brakes from T cells to enhance anti-tumor T-cell response.

The PD-1 molecule is another co-inhibitory molecule that is engaged by binding to its ligands PD-L1 or PD-L2 expressed widely on non-lymphoid tissues and some tumors, dampening T-cell activation in the periphery by downstream inhibition of the TCR-mediated activation. The development of antibodies blocking the PD1/PD-L1 axis has demonstrated efficacy in tumor rejection by reinvigoration of exhausted CD8 T cells [6]. Many other checkpoints are currently being targeted with antibodies in clinical development, either amplifying the activation of costimulatory immune checkpoints, or inhibiting other regulatory immune checkpoint molecules.

### 3.2. Epigenetic Modulations to Restore Anti-Tumor Immune Response

HDAC inhibitors have been firstly developed in oncology as anti-cancer drugs, because of the frequent overexpression of HDACs in several tumor types, however, it is important to underline that HDAC inhibitors can have a direct effect on immune cells themselves. Epigenetic regulations play pleiotropic effects implicated in the modulation of the immune system.

Regarding innate immunity, HDACs can act as both positive and negative regulators of Toll-like receptor (TLR) signaling [25]. Several TLR target genes are induced in an HDAC-dependent manner, including a number of genes that encode key inflammatory molecules, such as pro-inflammatory cytokines (interleukin (IL)-6, IL-12) or chemokines (CXCL10, CCL7, and CCL2). On the other hand, class I HDACs are implicated in the negative regulation of TLR signaling, promoters of inflammatory genes being directly repressed by HDACs [25].

HDACs seems to further play a role in adaptive immunity. Efficient anti-tumor immune response needs that cytotoxic CD8+ T cells recognize and kill cells displaying foreign antigens, like tumor neoantigens, via MHC class I (MHC-I) molecule’s presentation, and it has been shown that downregulation of antigen presentation and MHC-I expression are known resistance mechanisms to ICI [26]. In preclinical models, loss of histone acetylation has been described to be potentially implicated in the downregulation of MHC-I presentation via decreasing expression of transporter associated with antigen processing (TAP-1), a molecule involved in the transport of peptides from the cytosol to the endoplasmic reticulum implicated in the formation process of MHC-I peptide complex [27,28]. Using HDAC inhibitors, MHC-I presentation could be enhanced, leading to an increase in the expression of TAP-1 and TAP-2 and better tumor control in a mouse model of Merkel cell carcinoma [28].

DNA methylation can also regulate the expression of genes implicated in the process of antigen recognition, and a low dose of azacitidine has been shown to increase expression of antigen presentation molecules in several human cancer cell lines [29]. Moreover, the expression of tumor specific antigens by different ovarian cancer cell lines, such as cancer/testis antigens (CTAs), along with MHC-I molecules, can be induced by DNA hypomethylating agents, such as azacytidine, thus increasing the ability of cancer cells to be recognized by antigen-reactive CD8+ T cells [30].

Epigenetic remodeling also occurs during T-cell activation, differentiation, effector function, and exhaustion, via DNA methylation or HDACs regulation [31,32]. Preclinical data showed that the combination of HDAC and DNMT inhibitors in non-small cell lung cancer (NSCLC) cells correlated with increased interferon-alpha and beta signaling, upregulation of the antigen presentation machinery, and enhanced tumor control associated with increased T-cell infiltrate in the tumor microenvironment and reversion of T-cell exhaustion [33].

On the other hand, demethylation can lead to upregulation of inhibitory signaling leading to T-cell exhaustion, like upregulation of PD-L1 expression on NSCLC cells observed after treatment with azacitidine in vitro [34]. Same results were reported with the use of class I HDAC inhibitors on melanoma cell lines, inducing a rapid upregulation of histone acetylation of the PD-L1 gene, an enhanced expression of PD-L1, and an increased tumor control in combination with PD-1 blockade in a murine model, highlighting the high potential of epidrugs to sensitize patients to ICI.

Furthermore, the presence of immunosuppressive regulatory T-cells (Tregs) in the tumor microenvironment can inhibit effector T-cell and NK cell functions, leading to tumor immune escape. HDAC9 is essential to maintain the homeostasis of Tregs via direct interaction with FOXP3 gene, the major transcription factor mediating Tregs differentiation and suppressive functions [35]. In this case, administration of an HDAC inhibitor in vivo has led to increased FOXP3 gene expression and suppressive functions [35]. Conversely, using HDAC6-selective inhibitors induced a decrease in the frequency, FOXP3 expression, and suppressive function of Tregs on T-cells from melanoma patients treated in ex vivo cultures [36].

Another HDAC6-selective inhibitor, nexturastat, in combination with an anti-PD-1 agent in syngeneic melanoma tumor models demonstrated significantly reduced tumor growth and profound changes in the tumor microenvironment, such as reduction of pro-tumorigenic M2 macrophages along with an increase of effector T-cells [37].

The same findings were reported with SS-208, another HDAC6 inhibitor, showing impaired tumor growth in a syngeneic melanoma mouse model mediated by immune-related antitumor activity, as evidenced by increased infiltration of CD8+ and NK+ T cells along with enhanced M1 to M2 macrophages ratio in the tumor microenvironment [38].

Regarding myeloid-derived suppressor cells (MDSCs), data showed that HDAC11-deficient mice presented a highly suppressive immune infiltrate leading to more aggressive tumors as compared to wild type mice, suggesting the role of HDAC11 in MSDCs function [39]. However other data showed that the use of etinostat, a class I HDAC inhibitor, enhanced the antitumor effect of PD-1 inhibition in two syngeneic mouse tumor models. This positive therapeutic effect might be explained by the inhibition of MSDCs immunosuppressive functions [40].

These data show the complexity of the interactions between epigenetic regulation and the immune system, with epigenetic modulations that can have opposite effects on the function and migration of immunosuppressive cells. Additional data are needed to better understand these interactions. Development and use of selective HDAC inhibitors, targeting specific HDAC, might also help enhancing a more specific stimulation of anti-tumor immune response.

Figure 1 resumes the main interplay between epidrugs and anti-tumor immune response.

### 3.3. Clinical Evidence of HDAC Inhibitors and ICI Combination

With the rationale for combining HDAC inhibitors to ICI, extensive clinical research has been set up to assess these combinations in the setting of advanced solid tumors. Published data from early clinical trials are already available, showing that combinations of HADC inhibitors with ICI are feasible, with early efficacy results resumed in Table 1. The Table 2 resumes ongoing studies assessing the combination of HDAC inhibitors with immune checkpoint inhibitors in advanced solid tumors.

The first developed pan-HDAC inhibitor, vorinostat, was assessed in combination with pembrolizumab in patients with metastatic NSCLC in a phase I trial, showing a favorable safety profile with no dose-limiting toxicity and preliminary encouraging efficacy results with an ORR of 13% in the 30 evaluable patients, mainly pre-exposed to ICI [41]. A further randomized phase II study compared pembrolizumab single agent to pembrolizumab plus vorinostat in non-pretreated metastatic NSCLC patients with PD-L1 expression TPS > 1% [42]. The most common related adverse events included anorexia (43%), fatigue (43%), nausea (35%), and increased creatinine (35%) in the combination arm. Interestingly, patients with cold tumors (low pre-treatment tumor immune infiltration) in the combination arm showed an improved ORR of 66% as compared to the pembrolizumab alone arm (33%) with a DCR of 91% versus 58% (*p* = 0.017), encouraging further development of the combination.

A phase II study of vorinostat in combination with pembrolizumab and tamoxifen in heavily pre-treated HR-positive breast cancer patients (median of five prior lines of treatment) failed to show a substantial benefit of the combination and was halted prior to completion of target enrollment. Epigenetic modulation and priming of immunotherapy should be assessed in better selected patient populations, maybe in earlier settings of the disease [44].

In the ongoing PEVOsq basket trial, our team assesses the combination of pembrolizumab with vorinostat in patients with advanced SCC of six different tumor locations. Sequential biopsies aim at identifying predictive biomarkers of response to the combination, with collection of both mandatory blood and tissue samples at baseline, and further optional biopsies under treatment and at disease progression [56].

The ENCORE 601 trial was a phase Ib/II study designed to assess the efficacy of entinostat, a class-I HDAC inhibitor, in combination with pembrolizumab in advanced solid tumors, with phase 2 expansion cohorts in patients with advanced NSCLC and melanoma who have progressed after ICI treatment, and another expansion cohort in mismatch repair-proficient CRC patients. In recently published data of the ENCORE study in phase II expansion cohort of 71 evaluable NSCLC patients, the ORR was 9%, which did not meet the prespecified threshold for positivity [50]. In the expansion cohort of 53 melanoma patients the ORR was 19%, showing meaningful clinical activity in this particular population, with translational data showing a decrease in MSDCs and increased immune signatures in samples from responder patients [51].

Recent published data of a phase II study showed interesting results of entinostat in combination with nivolumab and ipilimumab in heavily pre-treated HER2-negative breast cancer patients (median prior therapies of 6, range 1 to 13), with an ORR of 30% among 20 evaluable patients, including a complete response in one TNBC patient [49], encouraging further clinical development. The authors described an increased ratio of CD8+/Tregs cells with treatment.

Entinostat was also assessed in a randomized phase II, placebo-controlled trial, in combination with atezolizumab versus atezolizumab plus placebo in patients with advanced TNBC patients, but showed disappointing results [46]. In this study, no significant difference in PFS was observed between the two groups, and a median OS of 9.8 months in the combination arm versus 12.4 months in the placebo arm. In addition, the combination arm resulted in higher frequency of severe adverse events, with a higher frequency of treatment-emergent adverse events leading to death. These data underline the complex interplays between epidrugs and ICI that will further need a deeper understanding, to better chose the ideal combination molecules and the specific patient populations who will benefit from the combination.

Mocetinostat, another class-I HDAC inhibitor, was assessed in several solid tumors. The preliminary results from a pilot phase I trial were recently published assessing mocetinostat in combination with nivolumab and ipilimumab in treatment-naïve metastatic melanoma patients [52]. In this study, among 10 treated patients, 2 patients experienced a complete response and 5 a partial response. However, all 10 patients had at least one grade 3 or 4 immune-related toxicity, limiting the development of the triple regimen. Interestingly, the authors showed ex vivo an accumulation of central memory CD8+ and CD4+ T-cells and decrease percentages and suppressive activity of MSDCs and Tregs in samples from blood of patients treated with mocetinostat [52].

Domatinostat, another class-I HDAC inhibitor was evaluated in combination with pembrolizumab in advanced melanoma patients with primary refractory or non-responding disease to prior ICI treatment [54]. Frequent adverse events related to domatinostat included diarrhea (23%), nausea (20%), fatigue (20%), rash (15%), pyrexia (13%), blood alkaline phosphatase increased (13%), vomiting (10%), dyspnea (10%), and all grade 1 and 2, except one grade 3 maculo-papular rash. No treatment-related death was reported nor increased rates of immune related toxicities. Among 40 treated patients, 1 complete response, 2 partial responses, and 9 disease stabilizations were reported with a clinical benefit rate of 30%, and 3 patients that were still on treatment 1.5+ years. The authors also showed a trend in a higher intra-tumoral expression of MHC genes and increased inflamed tumor microenvironment under treatment.

Interestingly, new types of immunotherapies are currently being evaluated in combination with HDAC inhibitors. For example, the phase I/II trial NCT0470847 is currently assessing entinostat with bintrafusp alpha, a bifunctional fusion protein targeting TGF-β and PD-L1 in combination with NSH-IL12, a recombinant human IL-12 heterodimer fused to a H-chain of the NHS76 antibody, a necrosis-targeting human IgG1 that will selectively deliver IL-12 to tumor.

Altogether, these data support further development of HDAC inhibitors in combination with immunotherapy agents in the context of specific solid tumors, even in patients who have previously progressed on anti-PD-1 or PD-L1 ICIs, so as to prime anti-tumor immune response and counteract tumor immune resistance mechanisms. These phase I/II studies can also give the possibility to accumulate translational data and improve understanding of the mechanism underlying interplays between epidrugs and immunotherapies.

## 4. Future Perspectives

Beside isoform-selective HDAC inhibitors, various new molecules, referred to as polypharmacological molecules, are being developed, exhibiting dual inhibition of HDACs with other therapeutic targets, usually used in the development of targeted therapies. For example, CUDC-101, currently assessed in phase I trials, is a novel small molecule which simultaneously inhibits HDAC, the receptor kinases epidermal growth factor receptor (EGFR) and human epidermal growth factor receptor 2 (HER2) in cancer cells [57].

Further development of epidrug plus immunotherapy combinations specifically uses epidrugs to prime chimeric antigen receptor (CAR) T-cells in the context of hematologic malignancies, for example, with either the DNMTi decitabine alone, chidamide alone, or the combination of both epidrugs (ClinicalTrials.gov Identifier: NCT04553393). The development of CAR-T cells is ongoing in solid tumors pending approval, and will certainly follow the same development process as in hematologic malignancies, applying improvements of technologies to solid tumors.

Another axis of development is to bring epidrugs earlier during the disease course, in order to help counteract multiple tumor resistance mechanisms acquired during time, in the locoregional setting, for example. This strategy is being investigated in stage III resectable melanoma patients (ClinicalTrials.gov Identifier: NCT04133948). In this study, a baseline biopsy is required to assess an interferon-gamma signature. Patients with a high signature will receive six weeks of domatinostat plus nivolumab before surgery, whereas patients with a low signature will receive the combination of domatinostat plus nivolumab and ipilimumab.

Large precision medicine trials, such as the IMPACT trial, could in the future help to better understand the interplays between epigenetic regulations and anti-tumor immune response. The investigators of the IMPACT trial recently analyzed data from next-generation sequencing and clinical records of 1661 pan-cancer patients from the IMPACT trial treated with ICIs to explore the association between lysine methyltransferase 2D (*KMT2D*) gene alteration and therapeutic efficacy of ICIs [58]. *KMT2D* gene encodes for a histone H3 lysine 4 methyltransferase, frequently mutated in cancer patients, with previous studies showing a correlation between *KMT2D* mutations and higher tumor mutational burden (TMB), suggesting that *KMT2D*-deficient tumors might be more sensitive to ICIs. In this study, patients with *KMT2D* alteration showed a significantly higher TMB than wild-type group (*p* < 0.01), along with a longer OS (median of 27 month versus 5 months, HR = 0.76, *p* = 0.02). The more frequent variant types of *KMT2D* were non-structural variants. Conversely, in the MSKCC-2017 cohort of patients who were not treated with immunotherapy, the OS of *KMT2D*-altered group was significantly shorter than the wild-type group (median OS of 22 months versus 26, HR = 1.20, *p* < 0.01), suggesting that *KMT2D* gene alteration might be an immunotherapy efficacy predictive factor, but not a prognostic factor. Development of new technologies to identify epigenetic biomarkers beyond methylation is ongoing.

## 5. Conclusions

Future development of immunotherapies requires the development of novel strategies to counteract tumor resistance mechanism and immune escape for patients who will not benefit from single agent ICIs. Epigenetics regulators, like HDAC inhibitors, can clearly be incorporated in combination strategies in this setting. Considering that HDAC inhibitors could have potential effect directly on tumor cells but also immune cells, these new molecules can be used to prime the immune system and improve anti-tumor immune response, becoming good partner drugs to combine with immunotherapy agents. Early phase clinical trials have already shown encouraging results with combination of HDAC inhibitors with immunotherapy for the treatment of solid tumors, actually moving forward to further clinical development. Many efforts are still needed to decipher interplays between epigenetic regulations, oncogenesis, and the immune system. Given the context-dependent function of epigenetic regulators, along with their pleiotropic effects, molecules like HDAC inhibitors may indeed induce very different effect from one to another situation. Further understanding of anti-tumor immune escape along with the involvement of epigenetic regulations will be needed to improve combination strategies of immunotherapies with epidrugs.

## Figures and Tables

**Figure 1 cancers-14-00066-f001:**
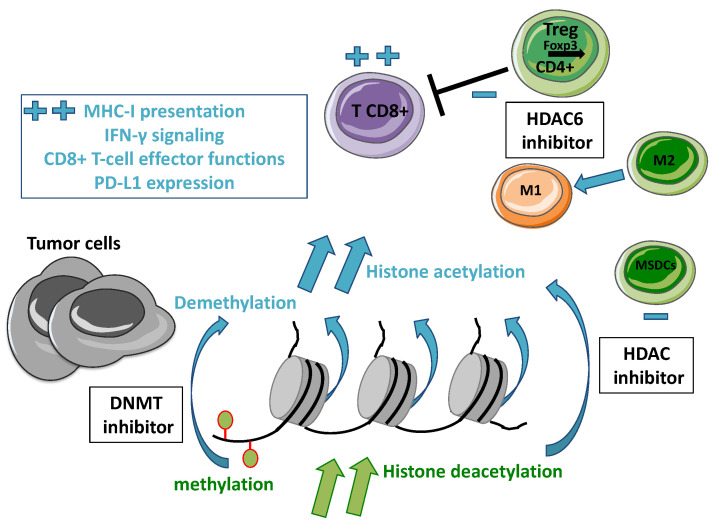
Mechanisms of epidrugs to restore anti-tumor immune response. Several mechanisms targeted by epidrugs to prime anti-tumor immune response are illustrated in this figure. Blue color represents the biological effects induced by HDAC inhibitors or DNMT inhibitors.

**Table 1 cancers-14-00066-t001:** Overview of clinical trials evaluating the combinations of HDAC inhibitors with immune checkpoint inhibitors in advanced solid tumors.

HDAC Inhibitor	Immunotherapy Agents	Other Drugs	Cancer Type(s)	Trial Phase	Efficacy Results	NCT Number
Vorinostat	Pembrolizumab		NSCLC	I/IB	ORR = 13% [41]	02638090
			PD-L1 TPS ≥ 1%	II	ORR = 48% [42]	
	Pembrolizumab		HNSCC, salivary gland tumors	II	HNSCC: ORR = 32% [43]	02538510
	Pembrolizumab	Tamoxifen	HR positive breast cancer	II	ORR = 4% [44] CBR = 19% [44]	02395627
Entinostat	Atezolizumab		HR positive breast cancer	I/II	ORR = 6.7% [45]	03280563

	Atezolizumab		TNBC	I/II	ORR = 10% [46]	02708680
					CBR = 37.5% [46]	
					PFS = 1.68 mo.	
	Atezolizumab	Bevacizumab	RCC	I/II	ORR = 20% [47]	03024437
					PFS = 7.6 mo.	
	Nivolumab + ipilimumab		Advanced solid tumors	I	ORR = 16% [48]	02453620
			HR+ and TNBC	II	ORR = 30% [49]	
	Pembrolizumab		NSCLC with previous PD under ICI	II	ORR = 9.2% [50]	02437136
	Pembrolizumab		Melanoma with previous PD under ICI	II	ORR = 19% [51]	02437136
Mocetinostat	Ipilimumab + Nivolumab		Melanoma	I	ORR = 70% [52]	03565406
Domatinostat	Avelumab		Mismatch repair proficient CRC, oesophagogastric	IIA	SD = 46% [53]	03812796
	Pembrolizumab		Melanoma with previous PD under ICI	Ib	CBR= 30% [54]	03278665
Romidepsin	Nivolumab	Cisplatin	TN or BRCA-mutated breast cancer	I/II	ORR = 44% [55]	02393794

NSCLC = non-small cell lung cancer; TPS = tumor proportion score; HNSCC = head and neck squamous cell carcinoma; HR = hormone receptor; TNBC = triple negative breast cancer; ORR = overall response rate; CBR = clinical benefit rate, corresponding to the addition of complete response (CR) + partial response (PR) + stable disease (SD); mo = months; RCC = renal cell carcinoma; CRC = colorectal cancer; and PD = progressive disease. NCT identifiers are available on ClinicalTrials.gov.

**Table 2 cancers-14-00066-t002:** Overview of ongoing clinical trials evaluating the combinations of HDAC inhibitors with immune checkpoint inhibitors in advanced solid tumors.

HDAC Inhibitor	Immunotherapy Agents	Other Drugs	Cancer Type(s)	Trial Phase	NCT Number
Vorinostat	Pembrolizumab		Renal or urothelial carcinoma	I/Ib	02619253
	Pembrolizumab		All types of SCC	II basket trial [56]	04357873
Entinostat	Pembrolizumab		Bladder cancer	II	03978624
	Pembrolizumab		Mismatch repair proficient CRC	II	02437136
	Avelumab		Ovarian cancer Advanced	I/II	02915523
	Bintrafusp Alpha + NHS-IL12		solid tumors	I/II	04708470
			HPV-refractory tumors		
Mocetinostat	Pembrolizumab	Guadecitabine	NSCLC	I/Ib	03220477
		(DNMTi)			
	Pembrolizumab		NSCLC	II	02954991
	Durvalumab		Advanced solid tumor and NSCLC	I/II	02805660
	Durvalumab		HNSCC	I	02993991
Chidamide	Toripalimab		Cervical cancer	I/II	04651127
	Nivolumab		NSCLC, RCC melanoma	I/II	02718066
	Envafolimab		NSCLC with previous PD under ICI	II	05068427
	Tirelizumab		Urothelial carcinoma	II	04562311
Panobinostat	Spartalizumab		NSCLC, CRC, TNBC	Ib	02890069
	Ipilimumab		Melanoma	I	02032810
Domatinostat	Nivolumab + Ipilimumab		Resectable muscle-invasive urothelial cancer	I	04871594
	Nivolumab + Ipilimumab		Stage III melanoma	I/II	04133948
Romidepsin	Pembrolizumab		Mismatch repair proficient CRC	I	02512172

NSCLC = non-small cell lung cancer; TPS = tumor proportion score; HNSCC = head and neck squamous cell carcinoma; HR = hormone receptor; TNBC = triple negative breast cancer; and CRC = colorectal cancer. NCT identifiers are available on ClinicalTrials.gov.

## Data Availability

The data presented in this study are available in the current manuscript.

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
