# Peer review of "HDAC Inhibition to Prime Immune Checkpoint Inhibitors"

_cancers, 2021, doi:10.3390/cancers14010066_

Round 1

Reviewer 1 Report

In this manuscript authors provide an update on recent discoveries on the use of epidrugs, and in particular of HDAC inhibitors, in combination with immunotherapies agents. The title and the abstract give the idea of a topic based on the latest findings in the use of HDACi in cancer immunotherapy, but at the end, in this manuscript, authors detailed only recent findings on the use of HDACi in combination with immune checkpoint inhibitors. However, cancer immunotherapy is not only based on the use of ICI but also of other recognized therapies which are not described in this manuscript, such as the use of cytokines, oncolytic viruses or Car-T cell therapy (only mentioned). Moreover, HDACi, used in combination with oncolytic virotherapy, have been shown to promote OV infection (Bartlett, D.L., Liu, Z.Q., Sathaiah, M., Ravindranathan, R., Guo, Z.B., He, Y.K., et al.Oncolytic viruses as therapeutic cancer vaccines. Mol. Cancer 2013.) or have been used in combination with IDO1 inhibitors (K. Fang, G. Dong, Y. Li, S. He, Y. Wu, S. Wu, W. Wang, C. Sheng, Discovery of novel indoleamine 2,3-dioxygenase 1 (Ido1) and histone deacetylase (HDAC)dual inhibitors, ACS Med. Chem. Lett. 9 (2018) 312-317). Thus, unless authors expand the topic to the effect of HDACi in the immune response and in relation to other immunotherapy agents, I would suggest to change title and abstract to be restricted only to the combination of HDACi with ICI. Anyway, I would still suggest to insert a paragraph which describes immune checkpoint signalling and its inhibition, I would put this paragraph at the beginning of section 3.

Moreover, at the beginning of section 2, I would put a short introduction describing epigenetics in general terms and show how it can affect cancer diseases.

Line 139: authors mention only vorinostat as approved for treatment of refractory cutaneous T-cell lymphoma but also Romidepsin has been approved for the treatment of cutaneous T-cell lymphoma and belinostat and Panobinostat have been approved for the treatment of peripheral T cell lymphoma and multiple myeloma, respectively. It is important that authors give an updated scenario of approved HDACi.

Section 3.1: here authors should also mention other studies using HDACi in combination with ICI such as S. Shen, M. Hadley, K. Ustinova, J. Pavlicek, T. Knox, S. Noonepalle,M.T. Tavares, C.A. Zimprich, G. Zhang, M.B. Robers, C. Ba_rinka, A.P. Kozikowski, A. Villagra, Discovery of a new isoxazole-3-hydroxamatebasedhistone deacetylase 6 inhibitor SS-208 with antitumor activity in syngeneic melanoma mouse models, J. Med. Chem. 62 (2019) 8557e8577.

Correct 2.2 title, line 102

Reviewer 2 Report

There are several questions and comments on the submissions:
1. Authors do not disclose the use of epigenetic therapy and immune checkpoint inhibitors in the treatment of pediatric brain tumors
2. Mechanisms underlying epigenetic therapy-immune system interaction should be described in more detail

Reviewer 3 Report

The review was aimed to describe the current view and the perspectives for clinical use of the inhibitors of histone deacetylases (HDACs) in combination with immune checkpoint inhibitors (ICIs). 

The manuscript is very-well written and illustrates the important role of HDAC inhibitors in regulation of the anti-tumor imune responses. The authors describe in detail the epigenetic alterations in cancer with the angle of histone modifications. The article also provides the molecular basis to use of epidrugs modulate the immune response and  therefore highlights the rationale for combining of HDAC s inhibitors with ICIs to treat the patients  with solid tumors. The ongoing clinical trials shown in Table 1 also support this view.  Despite that the results of several  trials shown in the manuscript were not encouraging and did not reach their primary points, this scientific approach looks very promising and might be used in future to overcome the tumors resistance to ICIs.   

Round 2

Reviewer 1 Report

Authors answered to all the raised questions, the manuscript is ready to be published